# On Saharan Air Layer Stability and Suppression of Convection over the Northern Tropical Atlantic: Case Study Analysis of a 2007 Dust Outflow Event

**Adrian Flores [1],\*, Ricardo K. Sakai [1], Everette Joseph [2], Nicholas R. Nalli [3] , Alexander Smirnov [4] , Belay Demoz [5] , Vernon R. Morris [6], Daniel Wolfe [7] and Sen Chiao [1]**

1   NOAA Center for Atmospheric Sciences and Meteorology, Howard University, Washington, DC 20009, USA
2   National Center for Atmospheric Research, Boulder, CO 80301, USA
3   NOAA/NESDIS/STAR, College Park, MD 20740, USA
4   NASA/GSFC/SSAI/BSB, Greenbelt, MD 20771, USA
5   Physics Department, University of Maryland Baltimore County, Baltimore, MD 21250, USA
6   School of Mathematical and Natural Sciences, Arizona State University, Tempe, AZ 85281, USA
7   NOAA/ESRL/PSD, Boulder, CO 80305, USA
\*   Correspondence: aflores@howard.edu

**Abstract:** A prominent Saharan Air Layer (SAL) was detected over the Northern Atlantic from the West African Coast to the Caribbean Sea in 2007. Data was collected from the Aerosols and Ocean Science Expedition (AEROSE), which encountered a major dust outflow on 13 and 14 May 2007. These observational measurements came from onboard instrumentation and radiosondes that captured the dust-front event from 13 to 14 May 2007. Aerosol backscatter was confined within the Marine Boundary Layer (MBL), with layers detected up to 3 km. Aerosol Optical Depth (AOD) increased by one order of magnitude during the dust front, from 0.1 to 1. Downward solar radiation was also attenuated by 200 W/m$^2$ and 100 W/m$^2$ on the first and second days, respectively. A weaker gradient at and above 500 m from potential temperature profiles indicates a less-defined MBL, and an ambient air temperature of 26 °C on 14 May and 28 °C on 15 May were observed above 500 m, reinforcing the temperature inversion and static stability of the SAL. Subsequent days, clear and boundary-layer cloudy days were observed after the dust front. From 14 to 18 May, a Convective Inhibition (CIN) layer started to form at the top of the MBL, developing into a negative buoyancy from 17 to 23 May, and reinforcing the large-scale anticyclonic atmospheric conditions. These results show that the SAL acts as positive feedback on suppressing deep convection over the tropical Atlantic during this dust outflow and several days after its passage.

**Keywords:** Saharan Air Layer; case study analysis; dust event; subtropical Atlantic

## 1. Introduction

The transport of Saharan dust minerals from the trade winds in the equatorial Atlantic is a prominent source of aerosols on the East Coast of the Americas [1–3]. The air mass associated with this transport, the Saharan Air Layer (SAL), transports aerosols and brings warmer and dryer air downwind [4]. The SAL and the Atlantic region have a significant role in the chemistry, physics, and thermodynamic properties that control the meso-synoptic weather processes that impact the Americas and are a prominent area of study. The impact of this region on large-scale dynamics and smaller-scale impacts in health and air pollution chemistry has been reported before [5,6]. Despite the large and important nature of the region, not much in situ data on the thermodynamics and aerosol properties exists except in the satellite-based passive remote sensing data sets. Recent studies of the SAL have used several data sources from different platforms. For example, satellite data [7,8], computational models [9–11], remote sensing [12], or a combination of datasets [13–16] were used.

There is a negative correlation between a tropical cyclone and the presence of dust aerosols in the tropical Atlantic and hurricane activity in the western Atlantic region [9,11,13,16]. The model dataset in Pan [10] revealed that the SAL is a prominent factor in the entire tropical Atlantic climate, provoking low static stability and warm air in the 950–500 hPa layer. These features are also shown in other studies [8,12,14,15]. Remote sensing data showed that the SAL depth and location change seasonally on the west side of the Atlantic basin, and it is less pronounced close to the African coast [7]. This includes the effect of telleconection patterns such as the North African Dipole Intensity (NAFDI) [17,18] on the variability of dust activity over the north subtropical Atlantic that in turn affects the seasonality, intensity, and spatial extent of the SAL. Using the NCEP/NCAR reanalysis data with a coarse vertical resolution, Wong and Dessler [19] suggested that the increase in temperature and decrease in humidity prevent the formation of deep convection over the tropical North Atlantic. Wong [20] also showed that the dust and dry anomalies of the SAL reduce thermal cooling at the top of the inversion layer and help to maintain the inversion layer through the Atlantic Ocean.

The purpose of this work is to present a case study to examine how the SAL can impact thermodynamic atmospheric conditions by suppressing deep convection and maintaining stability over the Atlantic Ocean. Previous studies are heavily based on numerical models and remote sensing data, which is mostly satellite data. The Aerosol and Ocean Science Expeditions (AEROSE) dataset [2,5] provides a great opportunity to observe the SAL using in situ measurements, thus providing a unique characterization of the impact of aerosol outflow that is of African origin on the thermodynamic parameters over the tropical/subtropical Atlantic Ocean. AEROSE is an experiment planned to address these science topics related to the role of African dust and smoke in atmospheric radiation and chemistry over the Tropical Atlantic by the U.S. National Oceanic and Atmospheric Administration (NOAA), in collaboration with the Howard University NOAA Center for Atmospheric Sciences and Meteorology (NCAS-M), as a series of long-term expeditions. This paper focuses on one of the AEROSE expeditions performed in 2007. The 2007 AEROSE cruise offers a unique case study of a major dust outflow over the Atlantic Ocean. Additionally, this campaign monitored this dust front before, during, and after its passage.

## 2. Instrumentation and Data Collection

AEROSE is a series of trans-Atlantic campaigns onboard the National Oceanic and Atmospheric Administration (NOAA) ship *Ronald H. Brown* where chemical and physical measurements were taken in situ and remotely during intensive observation periods [2,5]. Several instruments were on board the ship during these expeditions, and most of them continuously recorded measurements. The following paragraphs describe the instrumentation that has been used in this study.

Aerosol optical depth (AOD) measurements of 340 nm, 440 nm, 675 nm, 870 nm, and 936 nm were obtained using a hand-held Microtops II sunphotometer from Solar Light at Glenside, PA. The Microtops II parameters are set according to SIMBIOS (previous expeditions) recommendations. Each scan consists of 30 cloud-free scans every 30 min from 2 h after sunrise to 2 h before sunset. Then, data were processed following the Maritime Aerosol Network (MAN) protocol from the Aerosol Robotic Network (AERONET) [21]. Eighteen days of data were collected over the 26-day 2007 cruise.

A precision infrared radiometer (PIR) and a precision spectral pyranometer (PSP) from EPLAB in Newport, RI, were used to measure longwave and shortwave downward irradiance, respectively. The fluxes of these radiometers are considered for everyday measurements, especially to determine the cloudiness of the sky. These measurements were collected using a data logger, the CR3000, from Campbell Scientific at Logan, UT, which collected data each minute and generated a daily file. A CL31 ceilometer from Vaisala (Finland) was also on board the ship to obtain a continuous vertical profile of aerosol loading throughout the campaign. Measurements of aerosol extinction profiles were recorded every 15 s up to 7700 m at a resolution of 10 m. Radiosondes from Vaisala were used for

atmospheric profiling (e.g., pressure, temperature, humidity). About four launches were performed each day, resulting in 113 launches during the campaign. Data was also collected from the ship's instrumentation, such as sea surface temperature, air temperature, dew point temperature, and pressure. Details of the instrumentation are available at the ship's website (http://oceanexplorer.noaa.gov/technology/vessels/ronbrown/ronbrown.html, accessed on 11 March 2022).

### 3. Data Analysis

Figure 1 shows that there were three legs on the AEROSE 2007 cruise. The west-to-east transect (blue line) was the first leg (from Barbados on 3 May to 4 N 23 W on 11 May) of the cruise, followed by the second north–south leg (colored red; from 4 N 23 W on 11 May to 20 N 23 W on 18 May), and the third and final leg (colored yellow, from 20 N 23 W on 18 May to Fort Lauderdale, Florida, on 29 May).

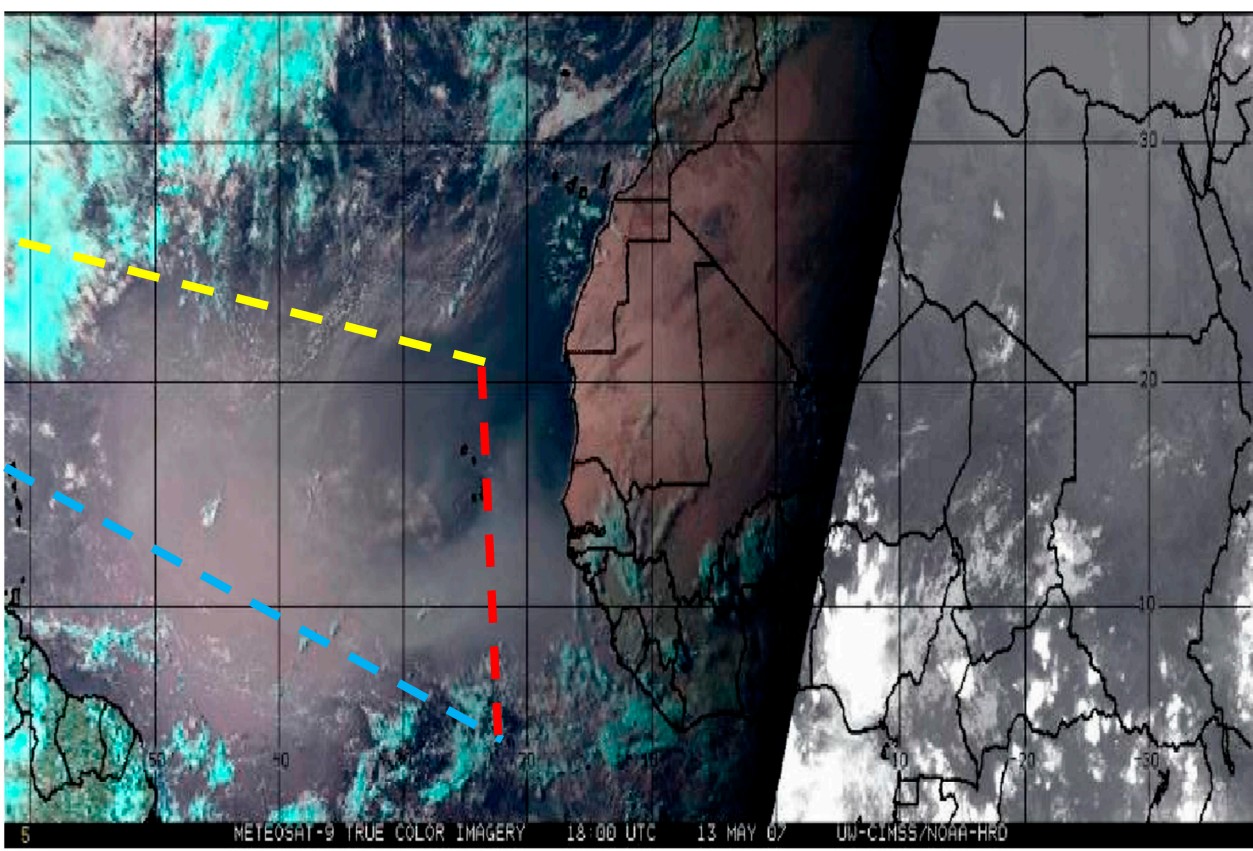

**Figure 1.** METEOSAT-9 true color imagery (13 May 2007) from the dust outflow event encountered during the expedition. Three lines depict the approximate cruise trajectory: blue from 3 May to 11 May; red from 11 May to 18 May; and yellow from 18 May to 29 May.

During the start of the first leg, conditions were calm, and with low dust concentrations and background AOD. However, by 7 May, an outflow of air associated with the Bermuda high started to bring some dust from the Saharan desert, which was an indicator that the ship was under the air mass that was associated with the SAL. A few days after starting the cruise, the ship entered the Intertropical Convection Zone (ITCZ) at an angle (northwest) for 2 days, 10 and 11 May.

On 11 May, the ship turned and headed northward, exited the ITCZ, and re-entered SAL conditions. The ship crossed the dust front into the main plume on its second leg (Figures 1 and 2). The MERRA-2 dust aerosol optical thickness product for 13 May (Figure 2) shows a major dust concentration over the West African coast with a northeasterly flow originating from the Sahara Desert. AOD data from Dakar's and Cape Verde's AERONET

(https://aeronet.gsfc.nasa.gov/new_web/data.html, accessed on 21 March 2022) show that the dust outflow passed over those stations on 9 May, as can be seen by the elevated AOD values, before encountering the dust plume on 13 May (Figure 3). The dust conditions lasted for 2 days (or the *Ronald H. Brown* was under a dust air mass for 2 days), as can be seen in the diminished downward solar radiation on 13 and 14 May. As the ship moved further north and out of the dust air mass, the air temperature, sea surface temperature, and dew point temperature decreased.

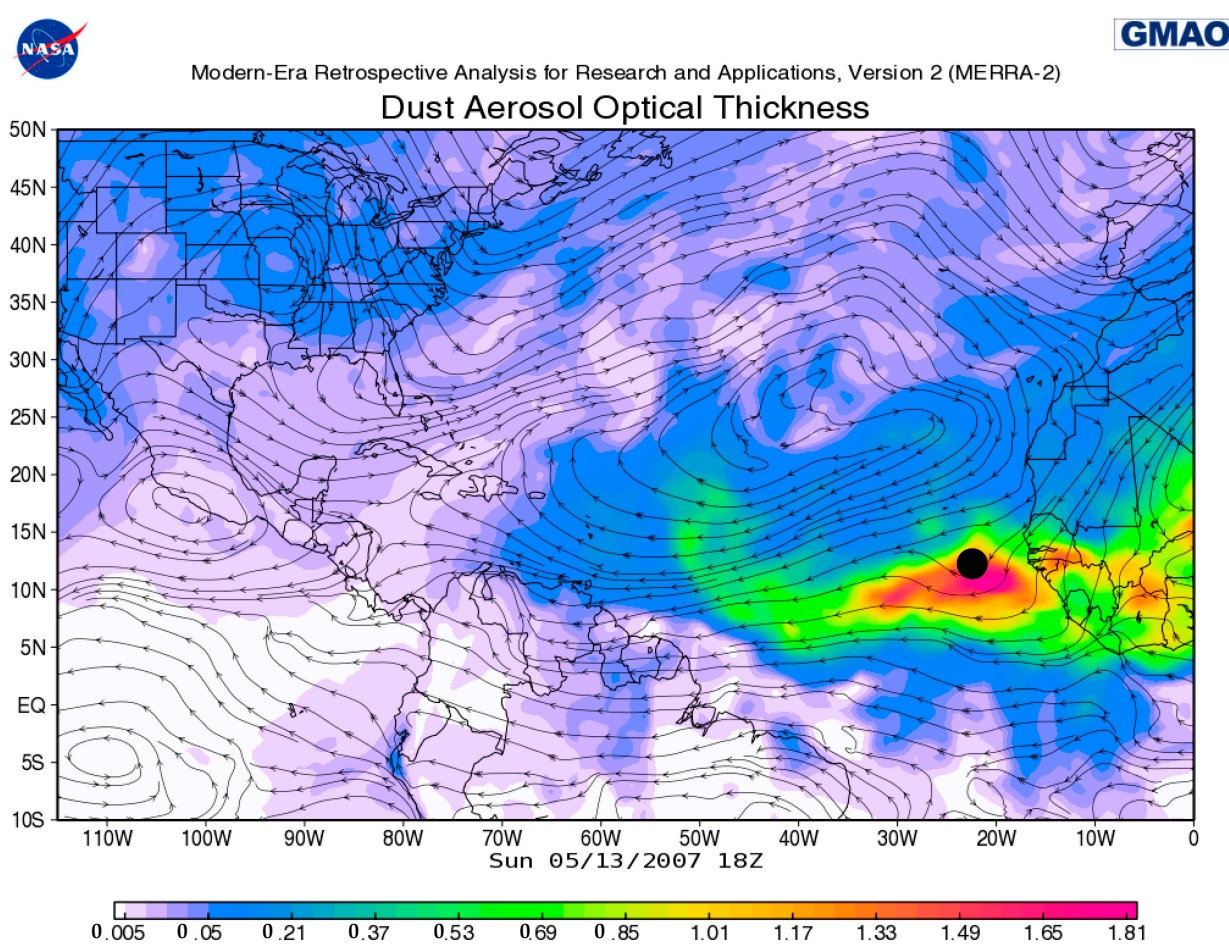

**Figure 2.** Dust aerosol optical thickness product from NASA's GMAO MERRA-2 for 13 May 2007. A black dot represents the approximate location of the ship.

The 440-nm and 870-nm channel AOD measurements from the Microtops-II were used to calculate the Angstrom Exponent (AE) because they were relatively free of major absorption bands (ozone, water vapor, and carbon monoxide). AOD and AE can then be used to classify the aerosol loading and particle size [22–24]. Figure 4 shows that the elevated AODs that were observed in this campaign were primarily coarse mode. Given the marine environment, the observed AE values of 0.1–0.2 were indicative of Saharan mineral dust aerosols. The highest AOD values measured corresponded to the time of the dust air mass passage over Dakar, Cape Verde, and the ship. Moderate AE values (greater than 0.5) probably resulted from smoke from biomass burning or man-made pollution. Negative AE values represent ice crystals in cirrus clouds. These clouds can pass the quality control for Microtops cloudless measurements. For example, on 9 and 21 May at the ship's location, high ceiling clouds were observed on the ceilometer (not shown).

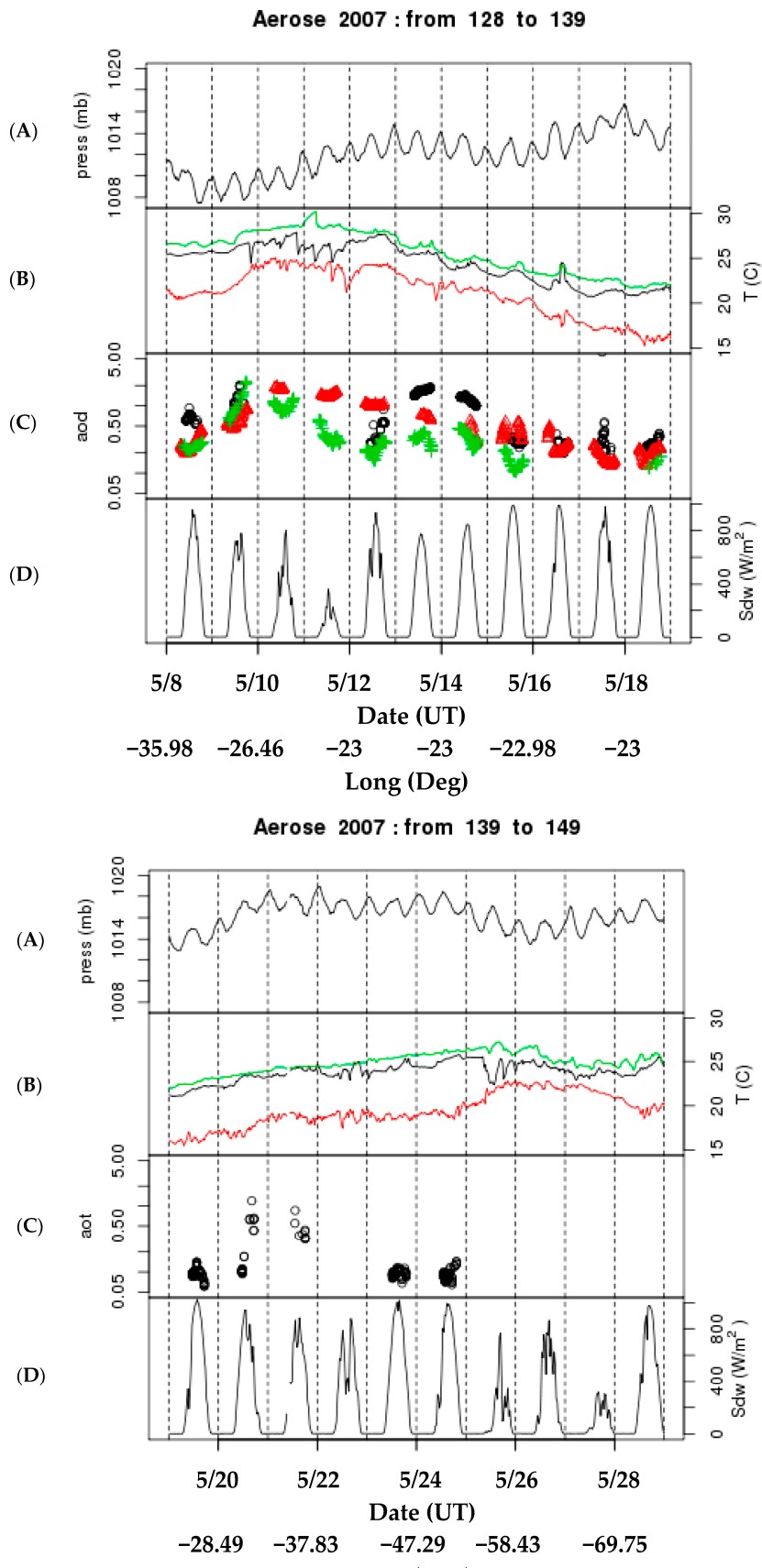

**Figure 3.** The first series of plots corresponds to the south to north leg, and the second series of plots to the east to west leg. (**A**) Surface (10 m) pressure measured at the ship mast, (**B**) ship-based surface

air temperature (10 m) (black), sea temperature (green), and dew point (red), (**C**) microtops AOD measured from the ship (black), Dakar (red), and Cape Verde (green), and (**D**) downward shortwave radiation measured from the Ronald H. Brown NOAA ship onboard the ship. No AOD measurements from Dakar and Cape Verde are introduced on the second series of plots because of the long distance.

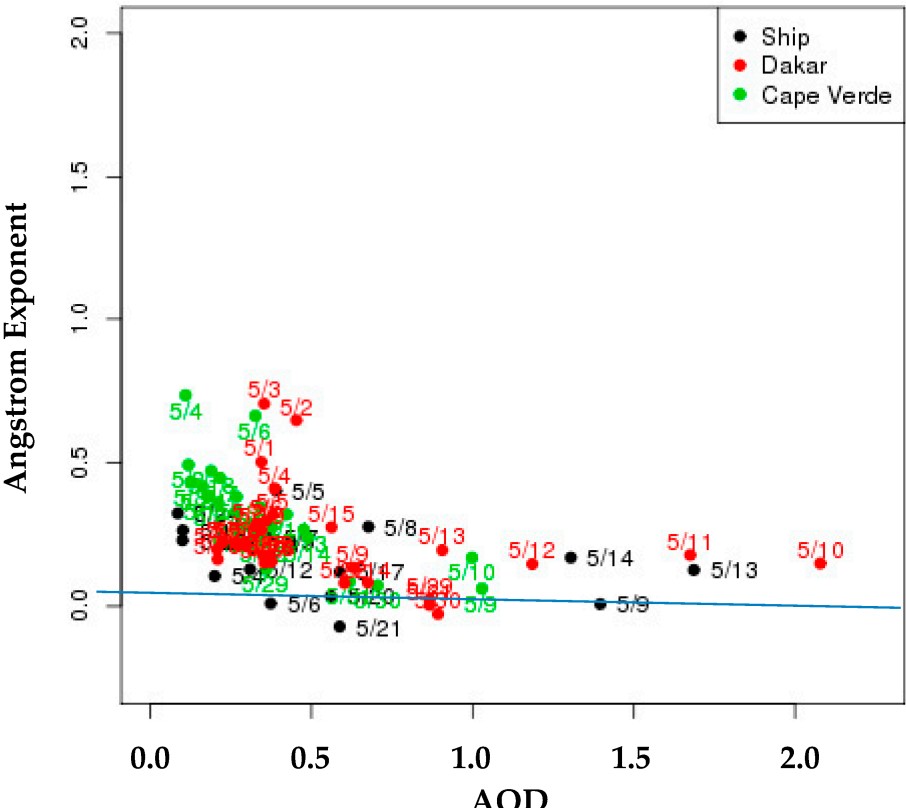

**Figure 4.** Daily averaged AE values versus AOD values from the ship (black), Dakar (red), and Cape Verde (green).

As shown in Figure 5, profile data on 13 and 14 May show a strong backscatter from the surface to above 3 km, which is consistent with a dust front crossing. Potential temperature shows that within the dust plume, there was a weaker gradient at about 500 m and above, indicating a less defined marine boundary layer (MBL). After 15 May, potential temperature gradients strengthened, indicating a strong cap at the top of the MBL (Figure 5). These gradients increased in height from 17 May to 23 May, detaching from the Lifting Condensation Level (LCL). A second mixing layer was formed between the LCL and the elevated gradient that eventually formed another cloud deck above the LCL/MBL height. On 19 May, when the ship started its third leg, aerosol backscatter signal values decreased between the LCL and the bottom level of the SAL, indicating clearer air. Backscatter values decreased above the sharp gradient of the potential temperature (where black lines agglomerate). The downward long-wave radiation bottom envelope started to decrease (Figure 5), such as the air and sea temperature patterns in Figure 3B. The spikes in the long-wave radiation are because of the presence of clouds. The air temperature heat map plot shows the contrast between the warmer SAL over the cooler air beneath it. Potential and air temperatures were similar from 13 to 23 May. On 14 and 15 May, local maximum air temperature profiles of 26 °C and 28 °C, respectively, were registered at about 500 m (Figure 6). A relative humidity (RH) heat map plot shows a dry tongue during the SAL event (Figure 6, green ellipsoid), and a sharp RH gradient is co-located with the potential temperatures (Figure 6, white lines).

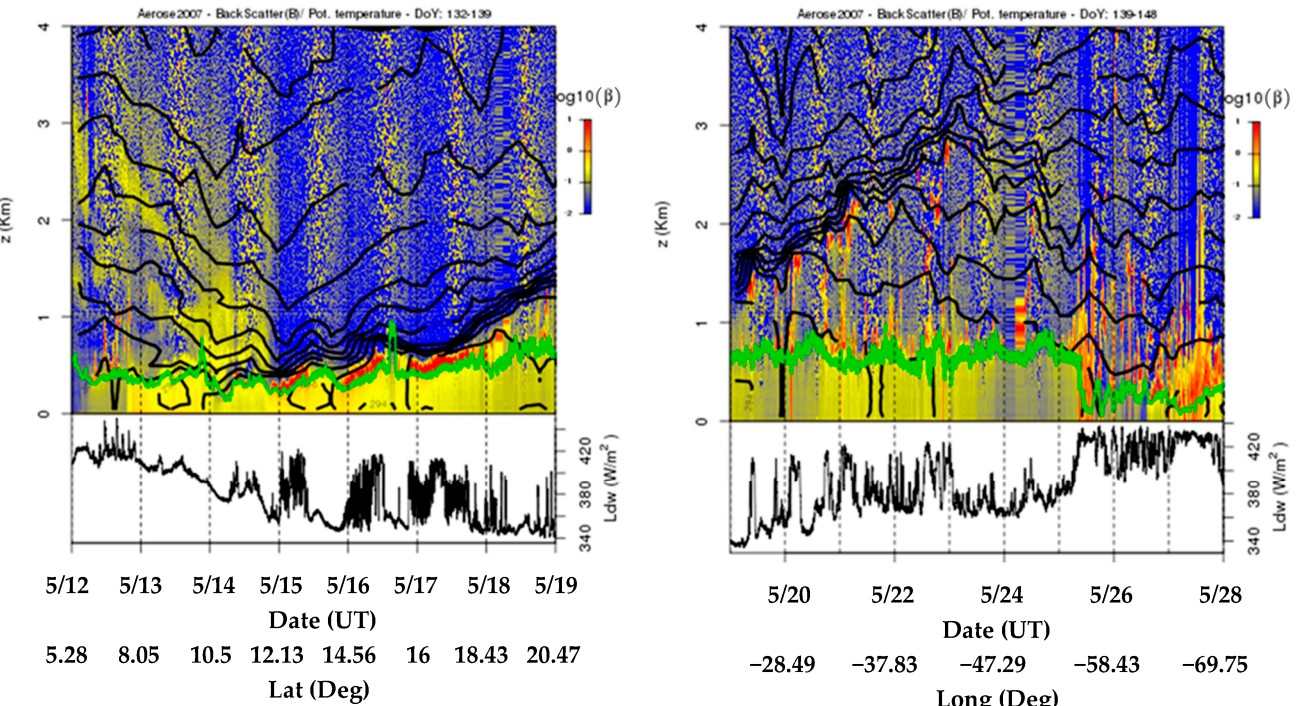

**Figure 5.** (**top**) Potential temperature contour plot (black lines) with backscattering and LCL location (green line), and (**bottom**) downward longwave radiation from the ship. The **left** plot corresponds to the 2nd leg (south to north) and the **right** plot corresponds to the 3rd leg (east to west) with dates and coordinates on the bottom.

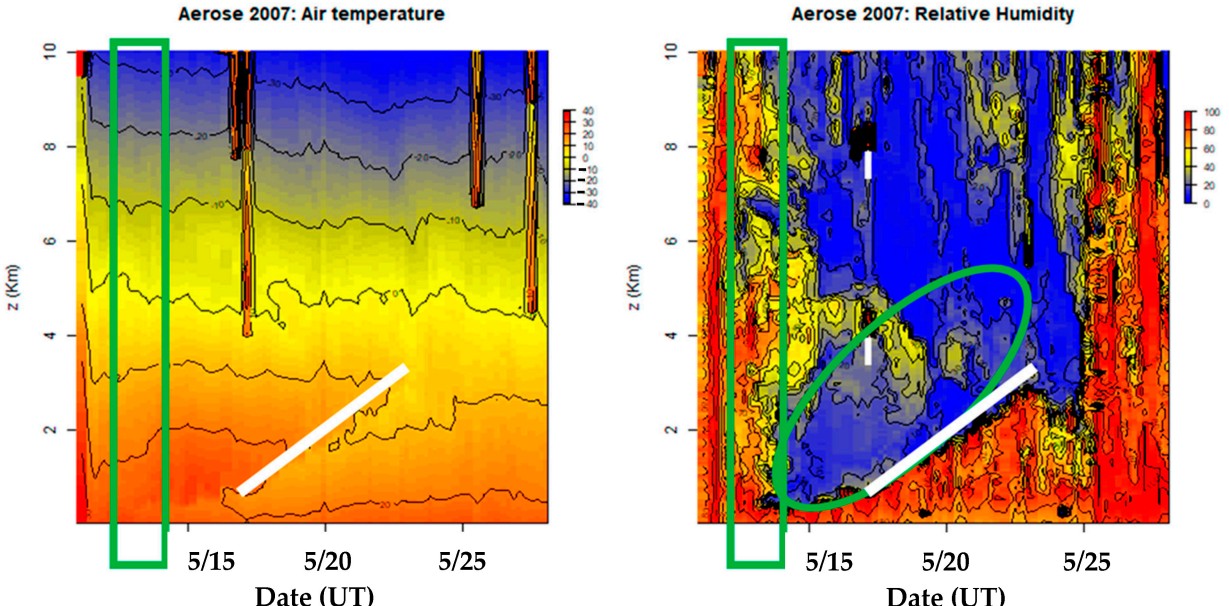

**Figure 6.** Heat map images for air temperature with black counter lines indicating different air temperature readings (**left**), and relative humidity with black counter lines showing dry and moist sections (**right**). The green rectangle represents the dust front. See text for white lines and an ellipsoid.

The yellow track that is highlighted in Figure 1 is the third and final leg of the expedition that started on 18 May 2007, and ended on 29 May 2007, and crossed the Atlantic Ocean from 20 N 23 W to Fort Lauderdale, Florida. During this leg, the dust event and SAL subsided, as seen in Figures 5 and 6. As the high pressure settled in, the air temperature and dew point gradually increased (Figure 3), suggesting that the ship had exited the SAL

environment. This third and final leg was associated with deep mixing, and persistent dry conditions that persisted from 19 to 24 May, followed by rain showers when it approached the Caribbean Sea.

## 4. Assessment of Data Analysis

The data analysis in this paper provides a cross-sectional schematic of a Saharan dust outflow event and its thermodynamic effects over the Atlantic Ocean. Figure 1 shows the great expansion of the dust plume exiting the Saharan desert and the SAL covering a vast area of the North Atlantic Ocean. With two days of dust loading and the surface temperature and solar radiation decreasing, Figures 3 and 4 show cross-sections of the dust event and its evolution over time and distance.

Upper air analysis from Figures 5 and 6 shows in great detail the effects of the dust plume over the Atlantic Ocean. All three plots show an undeveloped MBL before the dust front and a sharpening of its gradients on the following two days, 13 and 14 May. Post SAL days, the RH gradients became well organized, detailing an MBL that was defined by the LCL, climbing to a height of 1 km until the pattern was disrupted by precipitation on 25 May. The third leg was a continuation of the aftermath of the dust event, where the SAL's base kept increasing up to 3 km throughout the Atlantic Ocean until 23 May (Figures 5 and 6). Thus, the presence of SAL will reinforce the synoptic anticyclonic conditions (Figure 2) above the cloud layer for most of the second and third legs of the cruise. Surface temperature and downward longwave radiation started to decrease on 25 May (Figure 3), indicating that the ship had exited the SAL.

One of the important features of the dust event is the high temperatures above the MBL. Figure 6 shows an increasing air temperature of 26 °C and 28 °C above 500 m on 14 May and the beginning of 15 May, respectively. This local maximum could either be the result of heat advection from the Saharan desert and/or aerosol absorption just above the MBL from incoming solar radiation. This feature was observed during an earlier AEROSE campaign [25]. Dunion and Velden [4] also suggested that the temperature inversion helps to keep the SAL (stable/dry) intact well across the Atlantic Ocean. The Modern-Era Retrospective analysis for Research and Applications, Version 2 (MERRA-2), from NASA's Global Modeling and Assimilation Office (GMAO) 850 mb temperature and relative humidity May 2007 anomalies show an agreement of an increase in temperature and a decrease in relative humidity over the region of study (Figures 7 and 8, respectively). Studies have shown how to quantify the impact of SAL [26–28]. Figure 9 shows how 13 May, which was a major dust loading day, yielded average shortwave forcing values up to $-200$ W/m$^2$, resulting in a local cooling effect of 0.8 K/day. On 14 May, shortwave forcing values went up to an average of $-100$ W/m$^2$ with a cooling effect of 0.4 K/day.

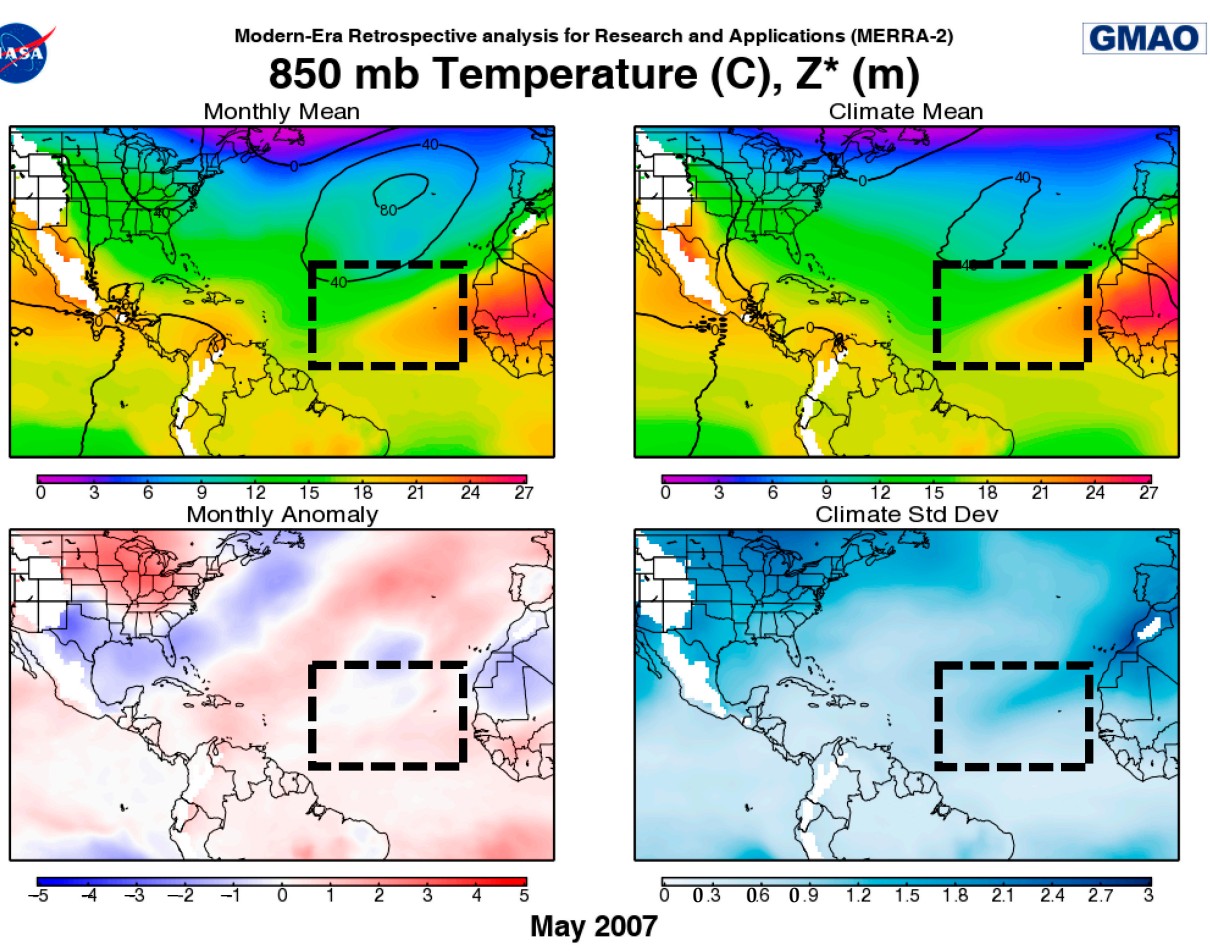

**Figure 7.** NASA's GMAO MERRA-2 850 mb temperature monthly product showing an increasing anomaly over the region of study (dash lines).

Convective studies also show how SAL can make an impact over the Atlantic [29–32]. Convective Available Potential Energy (CAPE) and Convective Inhibition (CIN) values were also calculated from each radiosonde sounding during the campaign (Figure 10). From 10 to 13 May, CAPE values were high and CIN values were close to zero, representing a pre-SAL marine tropical environment. Following this beginning on 14 May, the CAPE values plummeted to near zero, whereas the CIN values steadily increased to over 600. The CAPE and CIN values that were associated with the SAL were located from the MBL up to 4 km, as shown in Figure 8, are indicative of high static stability for these 2 days, and these conditions inhibited deep convection [19,25]. Thereafter, from 17 to 19 May, when the ship was traveling north, atmospheric conditions were still stable, with CAPE and CIN values that were close to zero, but most of the troposphere showed negative buoyancy (Figure 8). From 20 to 24 May, on the third and final leg of the journey, CIN values were, for the most part, greater than the relatively small CAPE values. Near the end of the campaign, after the ship had passed the SAL conditions, the CAPE values rebounded and increased substantially, while the CIN remained non-existent or zero, indicating a return to tropical convective conditions and the appearance of localized rainstorms.

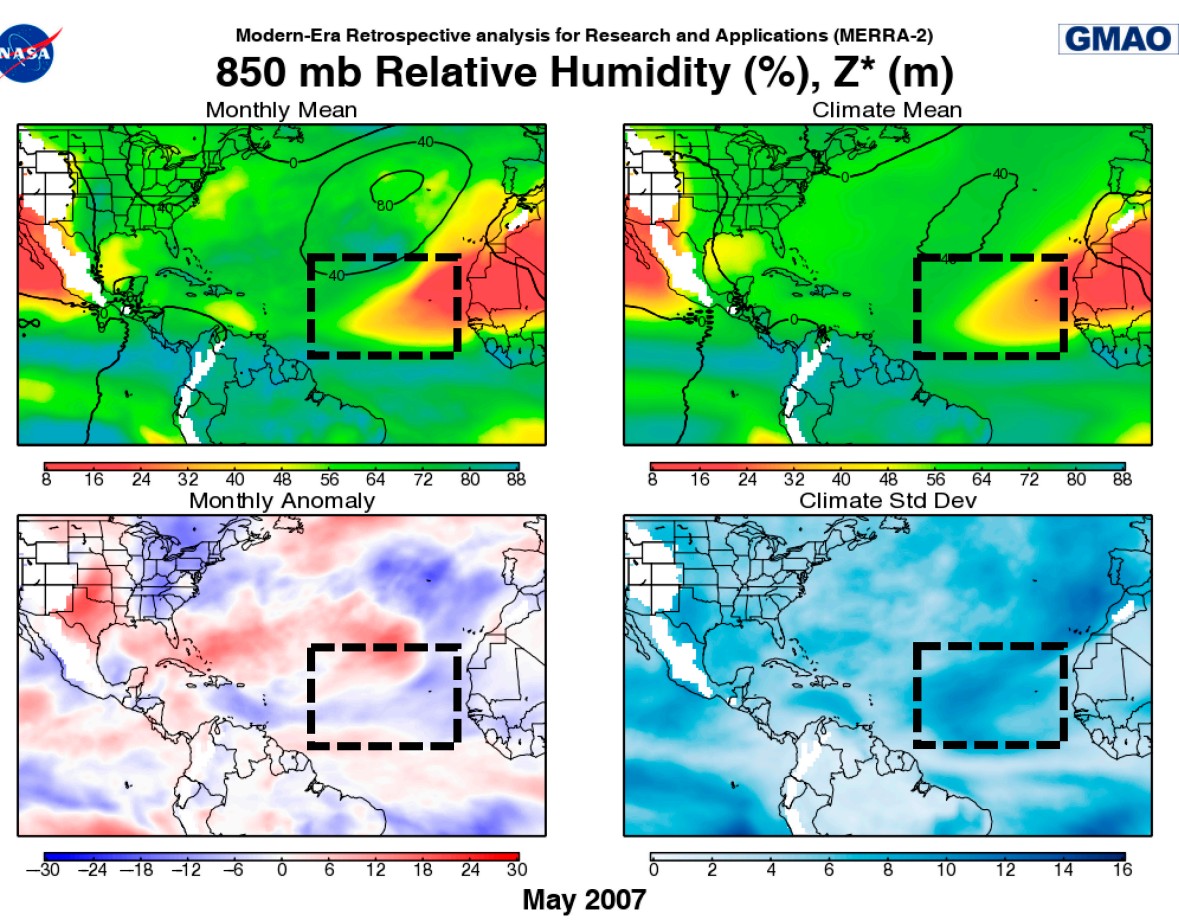

**Figure 8.** NASA's GMAO MERRA-2 850 mb relative humidity monthly product showing a decreasing anomaly over the region of study (dash lines).

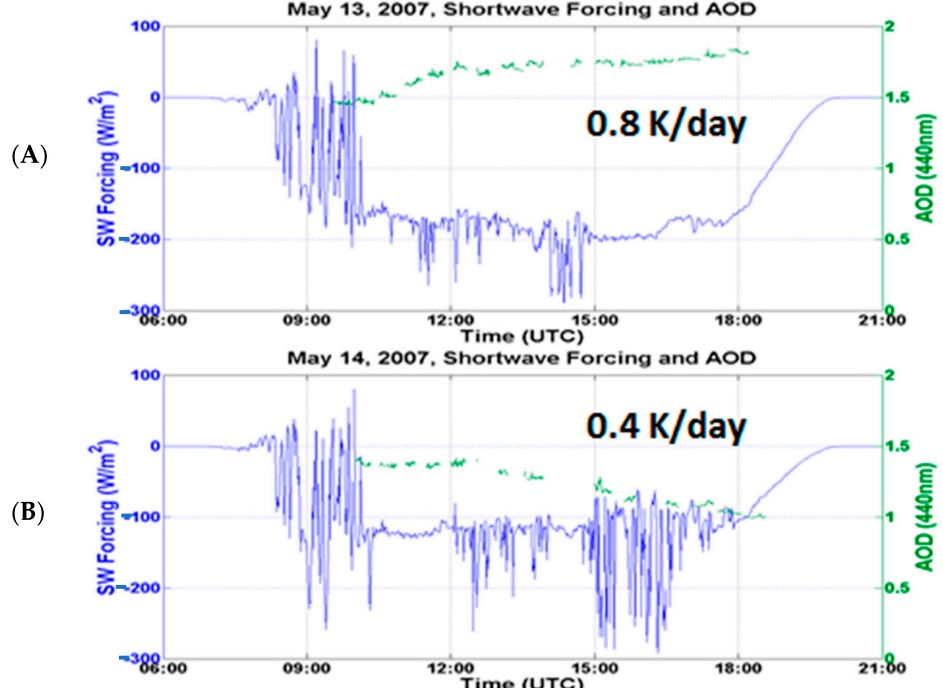

**Figure 9.** (**A**) Shortwave forcing (blue) and AOD (green) for 13 May 2007. (**B**) Shortwave forcing (blue) and AOD (green) for 14 May 2007.

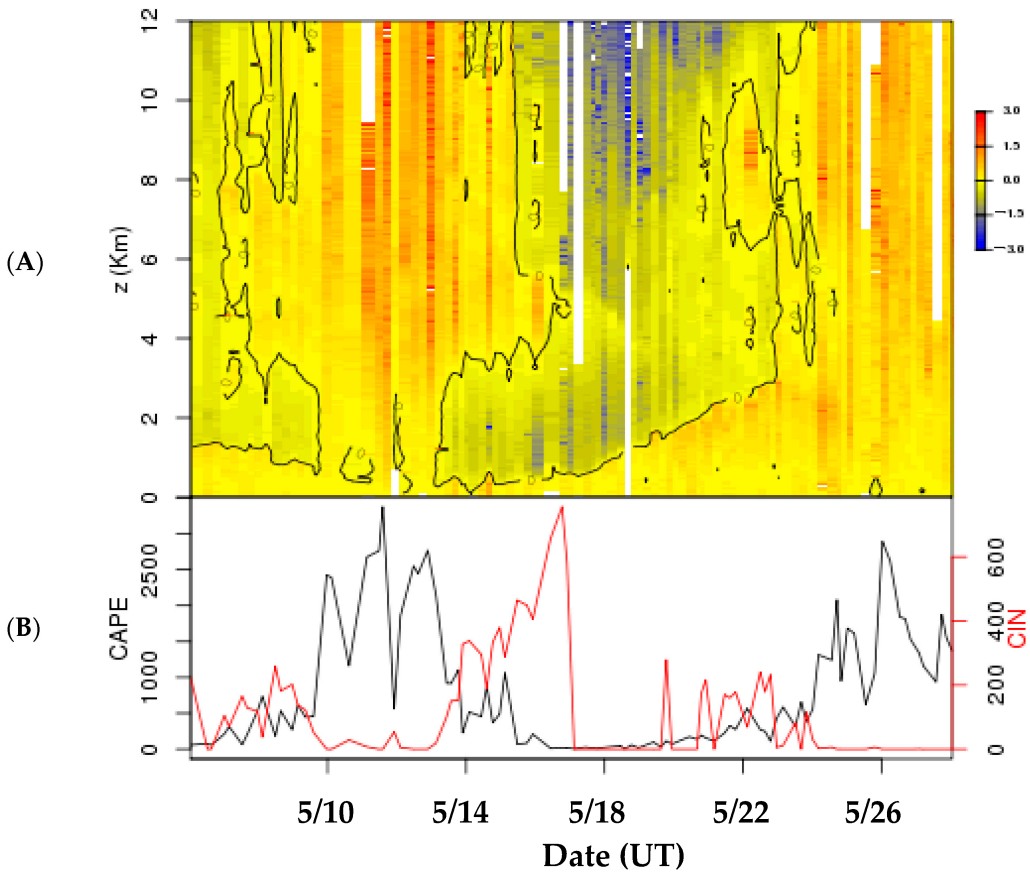

**Figure 10.** (**A**) Positive and negative buoyancy for the whole campaign, and (**B**) CAPE and CIN values from each radiosonde sounding obtained during the campaign.

### 5. Summary and Future Work

This study presents the impact of SAL on tropical Atlantic atmospheric conditions during a dust outflow event that was captured during the 2007 AEROSE campaign. A detailed spatial and temporal description of the SAL effects on the atmospheric conditions using observational measurements aboard a vessel crossing the tropical Atlantic was shown. Although the dust front was the most prominent feature for SAL detection using remote instruments, such as a Lidar and sun photometer, SAL effects lingered longer by impacting the thermodynamic profiles. A warm and dry air mass dominated the atmospheric conditions for most of the rest of the cruise. This air mass increased the stability that was already established from the Bermuda high over the North Atlantic by reinforcing the temperature inversion and creating a CIN layer above the trade wind cloud layer, which prevented deep convection throughout the meridional transect at the eastern part of the tropical Atlantic.

Data from this campaign and other AEROSE campaigns are used to validate the NOAA-Unique Combined Atmospheric Processing System (NUCAPS) sounding products derived from the Suomi National Polar-Orbiting Partnership (SNPP) satellite [33–35]. The data from recent and future campaigns will be used to compare this study and validate future products from NUCAPS.

**Author Contributions:** Conceptualization, R.K.S.; software, D.W.; formal analysis, A.S.; resources, V.R.M.; data curation, N.R.N.; writing—original draft preparation, A.F.; visualization, B.D.; project administration, E.J.; funding acquisition, S.C. All authors have read and agreed to the published version of the manuscript.

**Funding:** This research is based on work supported by the U.S. Department of Commerce, National Oceanic and Atmospheric Administration, Educational Partnership Program under Agreement No. NA22SEC4810015.

**Institutional Review Board Statement:** Not applicable.

**Informed Consent Statement:** Not applicable.

**Data Availability Statement:** Datasets analyzed during the current study are available by request as written in the NCAS-M website (http://ncas-m.org/research/data-management/, accessed on 21 August 2022).

**Conflicts of Interest:** The authors declare no conflict of interest.

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
