# Peer review of "On Saharan Air Layer Stability and Suppression of Convection over the Northern Tropical Atlantic: Case Study Analysis of a 2007 Dust Outflow Event"

_atmosphere, doi:10.3390/atmos14040707_

Round 1

Reviewer 1 Report

The subject of the paper clearly falls within the scope of this Journal.

The paper is very interesting, well written and well organized, and represents some advancement over the actual state-of-the-art. The ways and means are well described as well as the obtained results which are thoroughly discussed and conclusions are well drawn. The paper is also supported by some literature review. However, in order to make for a stronger paper, I suggest that the authors should cite and discuss the following relevant paper, which could be used as benchmark towards the proposed approach:

- Gomes et al., “Influence of an Extreme Saharan Dust Event on the Air Quality of the West Region of Portugal”, Gases, 2, 74–84 (2022) - DOI: 10.3390/gases2030005

I do recommend the publication of this paper, subjected to these changes.

Author Response

Thank you so much for your comments and taking the time to review this manuscript. Your recommendation has been taken in consideration and it is now part of the reference for this paper. Thank you so much for your suggestion.

Reviewer 2 Report

This paper examines the stability of Saharan dust air layer over the tropical/subtropical Atlantic during a dust outflow in May 2007. The analysis is based on ship-borne measurements during previous cruise campaigns in the Atlantic. Results show that the dusty layer originated from Sahara causes a series of meteorological dynamics above the marine boundary layer, leading to suppress of deep convection and reinforcing anticyclonic conditions. The paper is mostly meteorology-oriented, it's well written, but the figures used should be redrawn as they are not very clear for the reader. Also, the reference list should be improved with studies related to meteorology-dust dynamics over the marine environments. In the attached pdf file, I have included comments at specific parts of the text that authors may follow in a possible revision.    

Author Response

Thank you so much for your comments and taking the time to review this manuscript. The figures have been modified as recommended from your notes on the paper. The reference list has been populated with more articles, including the ones suggested on your review. Here are more line by line changes done on the paper:

Line 25: Changed to MBL.

Line 29: Added Subtropical Atlantic.

Line 49: Added the suggested references.

Line 75: Deleted "using".

Line 99: For some reason, the lines didn't come on the reviewing paper, but they have been added now.

Line 108: Again, sorry that this feature didn't come on the reviewing paper. It is now there.

Line 119: Deleted "solar".

Line 122: Made changes for better reading of the graphs.

Line 133: You're right, more papers need to be cited for this part and a few more where added to enrich this part of the study.

Line 137: You have a point. These are not high, but moderate. The wording is confusing, but it has been corrected now.

Line 149: Forgot to define Lifting Condensation Level. Thank you.

Line 165: Yes, it will be stated that the black contour lines are potential temperature contour lines.

Line 168: Thank you. It is now stating that the lines are air temperature contour lines.

Line 210: You're right. Figure 7 has been moved to section 3 as data analysis result.

Line 229: The same with figure 8, it is now in section 3 as data analysis result. A few more papers were cited for this analysis.

Reviewer 3 Report

In the manuscript the authors present experimental results related to the Saharan dust intrusion over Northern Atlantic obtained during the Aerosols and Ocean Science Expedition (AEROSE) in 2007. The measurements were conducted onboard and include both in-situ and remote sensing techniques. Three cruise trajectories were examined and the influence of dust on tropical meteorological conditions are presented.

Generally, this is interesting case study providing valuable experimental information since each such expedition is unique. The results might be interesting to the wider community, but it seems that the manuscript is not carefully prepared and should be moderately improved.

 Please find below several suggestions that might be considered:

 -        Having in mind that the results are obtained almost 15 years ago, from my point of view some recently finding available in literature should be included and commented. In introduction part more information on Saharan dust intrusion over Atlantic should be explicitly mentioned, and most important previous studies and challenges should be analyzed

-  Discussion section should be slightly expanded in terms of similar (experimental or modeling) results available in literature

-        It is stated that the data are processed using for example AERONET protocol from 2009. Is there a new protocol for data processing and analysis? What kind of analysis was used for ceilometer backscatter coefficient calculation? Obviously, there is significant improvements in algorithms used in the last decade. Did the authors use reanalysis?

-        Any kind of measurements should include error analysis. I would strongly suggest to include error bars and analysis in figures. This is important since the main contribution of the paper is related to these experimental measurements

-        It would be useful to find if there are some other satellite, modeling or other available data that might be included in Supp. Mat. showing dust transportation.

Minor suggestions:

All figures should be improved in terms of quality, legends and axis visibility, units, fonts, panels notations…

Figure 1: It seems three colored lines showing legs are not visible

Figure 3: Notations of panels are not appropriate. Please check quality of figure

Author Response

Thank you so much for your comments and taking the time to review this manuscript. We have adopted your suggestions and here are the actions that we took part by part on your comments:

  • Yes, the campaign took a long time ago and more articles have been published ever since. So, some new references were added to enrich the publication like Rodriguez et al. 2015 and Cuevas et al. 2017. More information has been to the introduction on how the SAL affects the thermodynamics of the Atlantic. And, the introduction will also cover previous studies, like SIMBIOS, and the challenges, but not leaving out the uniqueness this study presents.
  • The discussion section has been modified and extended to reflect the experimental results discussed in other articles like Nalli et al. 2011 and Kanitz et al. 2013.
  • Data processing and analysis protocol remains the same as described on NASA's Maritime AERONET website (https://aeronet.gsfc.nasa.gov/new_web/man_data.html). So, we're following their method and publishing our data on their website. The ceilometer backscattering calculation is an averaged of every 10 to 100 meters of the profile from each measurement. The ceilometer plots are actually an improvement from the original plots, that's where we get a better look of the MBL. There was a reanalysis of the data presented on this paper for its publication. All this discussion is added to the manuscript.
  • Thank you for the suggestion to include error bars. Because we're showing a few days of instantaneous measurements and no modeling, we have decided not to include error bars, but the quality control of the data reflects the effects of the dust event on the atmospheric conditions over the Atlantic, like temperature and relative humidity.
  • NASA's GMAO MERRA-2 has been added to the manuscript to show the dust transportation for May 13, 2007.

We did notice that some of the figures didn't come out like we submitted it originally. We corrected the mistakes, including figure 1 and 3, and enhanced the labels for better reading. Thank you so much for your input.

Reviewer 4 Report

This is a very short paper, using old data (2007) of Saharan dust measurements above North Atlantic form a ship expedition. In fact, there is nothing new in the paper, since there are already many papers on the subject. The authors only described a case-study. Also, how can the authors write in the abstract “These results show that the SAL is a positive feedback to suppress deep convection over the tropical Atlantic during the dust outflow and several days after its passage” since their analysis is based on one case only?

Thus, the paper is far from the publication standards of “Atmosphere”. The authors can strongly improve the paper by considering other case-studies, or by extending their analysis to comparison and discussion with others works on the same subject.

Specific comments:

The coloured lines are not visible on Figure 1.

Line 102: Figure 2 do not present what is said in this sentence.

Figure 2: The yellow star is not visible. Also, there is no indication of the coloured symbols plotted on the figure.

Figure 3: What is the meaning of the titles “fromv128 to 139” and “from 139 to 149”?

Figure 3: Bottom legends are almost impossible to read.

Figure 4: The authors speak of Angstrom Exponet”, but give in the y-axis legend “Angstrom Coeff.”.

Line 149: The authors must define “LCL”.

Line 193-194: This sentence is unnecessary.

The conclusion is more a short summary than a real conclusion. Future perspectives should be given

Author Response

Thank you so much for your comments and taking the time to review this manuscript.

It is a technical paper that is only covering a case study, but the uniqueness of this paper is that is presenting in-situ data with a major outflow of dust where measurements were done over seas and not over land. So, we understand is very short with old data, but we see that this manuscript can prove some of the literature that covers the same topic and that's including satellite and modeling. The wording of the abstract is corrected to state that in this case study we see that there is a positive feedback from SAL on suppressing deep convection over the tropical Atlantic during this dust outflow.

We have added more references to validate this case study with other papers so we can compare and discuss the effects of SAL on suppressing deep convection. This topic can be enhanced with more in-situ measurements over the Atlantic and this is where our scientific campaigns play a huge role with data collection.

For some reason the lines in figure 1 didn't come up in the map like we submitted. We have corrected this error and added the lines.

Line 102: The sentence is only describing the conditions during the first leg. Figure 2 is showing what happened on May 13, 2007, during the second leg. It has been modified to state this more clearly.

Figure 2: The yellow star has been added to the map and explained the other symbols on the map.

Figure 3: These are the Julian days, but these have been substituted with "5/8 to 5/18" and "5/19 to 5/28", respectively.

Figure 3: The bottom legends have been made bigger for better read.

Figure 4: The y-axis has been changed to Angstrom Exponent.

Line 149: The Lifting Condensation Level has been added.

Line 193-194: Sentence has been deleted. Indeed, this statement was not needed here.

The conclusion was changed to summary and future work was added of applying this research to modeling.

Round 2

Reviewer 1 Report

The authors have successfully ammended the paper according to the suggestions of the reviewers. I do recommend that the paper is accepted.

Author Response

Thanks again for taking your time to review and approve the manuscript.

Reviewer 2 Report

Authors provided some few changes and additions in the revised manuscript (yellow highlights), following reviewer's comments, but the revised manuscript has not been significantly improved in view of physical explanations of the results, and connection with the literature of dust storm dynamics over the western Sahara and the exposure of dust over the subtropical Atlantic. According to my opinion, authors may analyse and discuss more dust-storm dynamics and related meteorology over the subtropical Atlantic, as well as the changes in the synoptic situation (NAFDI Index) that highly control the dust emissions.   

Author Response

My sincere apologies. It was my fault that I uploaded the wrong manuscript and only had minor corrections. I've uploaded the correct and improve manuscript with all the revisions taken care of, including the figures.

Please let us know if there's anything we left out. Again, sorry for the confusion. Thank you so much for your patience and time on reviewing the paper.

Reviewer 3 Report

The revised version of the manuscript is slightly improved, the authors addressed some of the previous comments. Although it is stated that the introduction section is improved with more information and references, I really cannot see it (only one reference was added).

The authors mentioned “…new references were added to enrich the publication like Rodriguez et al. 2015 and Cuevas et al. 2017”. There are no new references! Limited information is added, but extended discussion is omitted. Quality of the figures is still questionable. Since the measurements performed are major contribution of the paper at least some error assessment should definitely be mentioned.

Generally, I’m not against publishing, but paper is not carefully prepared and only slightly improved.

Author Response

(The authors gave the same response as above.)

Reviewer 4 Report

The authors have made just little changes, and have added 2 new interesting figures. They said that more references are added; in fact only one has been added. They said that they have changed the conclusion, but I cannot see such changes! Thus, if the authors can considered these two remarks, a new improved version paper should be acceptable for publication.

Author Response

(The authors gave the same response as above.)
